# A Comparative Study of Nine SARS-CoV-2 IgG Lateral Flow Assays Using Both Post-Infection and Post-Vaccination Samples

**DOI:** 10.3390/jcm11082100

**Published:** 2022-04-08

**Authors:** Leontine Mulder, Benoit Carrères, Franco Muggli, Alix Zollinger, John Corthésy, Adrianne Klijn, Giuseppe Togni

**Affiliations:** 1Clinical Laboratory, Medlon BV, P.O. Box 50000, 7500 KA Enschede, The Netherlands; l.mulder@medlon.nl; 2Clinical Chemistry, Ziekenhuis Groep Twente, P.O. Box 7600, 7600 SZ Almelo, The Netherlands; 3Société de Produits Nestlé S.A., Nestlé Research, Route du Jorat 57, 1000 Lausanne, Switzerland; benoit.carreres@rd.nestle.com (B.C.); alix.zollinger@rd.nestle.com (A.Z.); john.corthesy@rd.nestle.com (J.C.); adrianne.klijn@rdls.nestle.com (A.K.); 4Faculty of Biomedical Science, Università della Svizzera Italiana, 6900 Lugano, Switzerland; fmuggli@bluewin.ch; 5Unilabs Central Laboratory, 1296 Coppet, Switzerland

**Keywords:** SARS-CoV-2, COVID-19, lateral flow assay (LFA), post-vaccination, post-infection, mRNA-vaccine, inactivated virus vaccine

## Abstract

Background: Since the SARS-CoV-2 pandemic, lateral flow assays (LFA) detecting specific antibodies have entered the market in abundance. Despite being CE-IVD-labeled, the antigenic compounds of the assays are often unknown, the performance characteristics provided by the manufacturer are often incomplete, and the samples used to obtain the data are not detailed. Objective: To perform a comparative evaluation of nine lateral flow assays to detect IgG responses against SARS-CoV-2. For the evaluation, a carefully designed serum panel containing post-infection samples and post-vaccination (both mRNA vaccine and inactivated virus vaccine) samples was used. Results: The sensitivity of the assays overall ranged from 9 to 90.3% and the specificity ranged from 94.2 to 100%. Spike protein-containing assays performed generally better than the assays with only nucleocapsid protein. The sensitivity of some assays was higher on post-infection samples, while other assays had a higher sensitivity to post-vaccination samples. Conclusion: A comparative approach in the verification of LFAs with an adequately designed serum panel enabled the identification of the antigens used in the assays. Sensitivities differed between post-infection and post-vaccination samples, depending on the assays used. This demonstrates that the verification of assays must be performed with samples representative of the intended use of the assay.

## 1. Introduction

A novel coronavirus first identified in Wuhan, China, in December 2019 has led to a pandemic, eliciting unprecedented global measures [1]. The clinical spectrum of COVID-19 ranges from asymptomatic infections and mild upper respiratory tract illnesses in the majority of patients to severe viral pneumonia with respiratory failure, multiorgan failure, and death [2]. Coronaviruses contain four structural proteins: the spike, envelope, membrane, and nucleocapsid proteins. The spike surface glycoprotein contains the receptor binding domain (RBD), which binds strongly to human ACE2 receptors and plays a major role in viral attachment, the fusion of viral and host membranes, and the entry of the virus into the host [3].

Testing strategies, in addition to control and treatment strategies, are crucial in fighting the pandemic. The early detection of the virus in both the symptomatic and the asymptomatic population is critical to accurately identify and isolate cases in an attempt to limit the spread of the virus. Methods based on molecular principles or antigen testing are mostly used for this purpose [4]. At the start of the pandemic, serological tests played an important role in epidemiological knowledge. However, for a diagnosis of active infection, there is no role for serological tests other than for the post-hoc diagnosis of individuals that did not receive a molecular test or had a negative molecular test with ongoing high clinical suspicion of SARS-CoV-2 infection. In some countries, a positive serological test can be used to prove past infection and obtain a pass to participate in society.

To date, there is still no consensus on the use of serology, and the proposed protection thresholds vary according to the study. Nevertheless, it is important to deepen the knowledge of all analytical tools used in order to define diagnostic strategies adapted to any epidemiological, logistical, and economic scenario.

Before the appearance of antigen tests, lateral flow tests (LFAs) allowed serological tests to be performed at point of care (POC) quickly and inexpensively. However, they must meet certain quality criteria—e.g., having a specificity >98% and a sensitivity >95% [5]. There are a huge number of LFAs, and their performance is generally characterized by the supplier itself using panels of limited and poorly described samples. In addition, the technical characteristics are not systematically shared with full details. During a pandemic, the procedure followed to approve new methods is generally shortened compared to the normal validation procedure. Governmental agencies, such as the FDA, responded to the coronavirus pandemic by allowing the Emergency Use Authorization (EUA) [6] to accelerate market accessibility. It allowed a multitude of tests to become available with variable specifications, including some with poor performances due to insufficient specificities/sensitivities, or because they were not representative of the population. The choice of samples used to precisely determine these two parameters is critical and must be sufficiently large and diversified. The simultaneous analysis of multiple LFAs with the same serum panels will help to make the comparison more reliable.

In the literature, several studies can be found concerning SARS-CoV-2 LFAs [5,7,8,9]. The characterization of their performance is mostly conducted using well-defined post-infection sera and the results are either compared with ELISA, CLIA, or other assays. Although these studies show great heterogeneity, some comparing one or two LFAs and others comparing twenty-two, the overall conclusion of these studies is that the sensitivity of the LFA is lower in comparison with immuno-assays. For post-infection samples, most studies find a sensitivity between 70 and 90% with more outliers down than up. A meta-analysis reviewing 151 articles on SARS-CoV-2 serological assays found one LFA fulfilling the requirement of a sensitivity of at least 95% and specificity of at least 98% for samples from patients at least 14 days post onset of infection [5]. Few studies have included samples from vaccinated subjects instead of infected subjects [10,11].

In this study, we determined the performance of nine commercially available SARS-CoV-2 lateral flow assays. We selected four tests that were available in Europe and previously evaluated by one of the authors (Dr. Giuseppe Togni) for his activity in the company Unilabs SA (Switzerland) and randomly selected five from a large panel of available tests. All tests were analyzed using a sample panel consisting of 219 samples characterized as positive (reactive with anti-spike and anti-nucleocapsid IgG ELISA tests for infected patients and reactive with anti-spike IgG ELISA test for vaccinated patients) and 69 samples characterized as negative (non-reactive with ELISA tests). This allowed for a direct comparison of the different characteristics of the tests. Taken together, our study highlights the need to standardize the comparative approach and shows that the verification of these assays in the context of their intended use is needed.

## 2. Materials and Methods

**Cohorts**: Samples (taken between 5 May 2020 and 12 July 2021) from five patient cohorts were used for this study. The first panel was from patients who were followed after a primary COVID-19 infection (35 patients). A second panel was from health personnel who wished to participate in a serological control study (32 positive and 69 negative samples defined following the criteria described below). The third panel included people who took part in a seroprevalence study in the Italian Graubünden (36 patients). The other two panels were from vaccinated patients, which included one panel of patients vaccinated with mRNA vaccines (Comirnaty COVID-19 from Pfizer (22 patients) and Spikevax from Moderna (30 patients)), and a panel of patients vaccinated with inactivated virus vaccine (Sinopharm BIBP COVID-19 from Beijing Institute of Biological Products-Sinopharm-China National Biotec Group Co (64 patients)). Characteristics of the samples are given in Table 1.

The choice of samples was made to compose a panel with potentially positive samples from both infected and vaccinated people, supplemented with negative samples. All the samples were first analyzed with two immuno-assay tests (Euroimmun ELISA and Abbott Architect CMIA; see description below). The criterion for determining whether a patient had been infected (positive sample) was as follows: reactive with anti-spike and anti-nucleocapsid IgG ELISA tests. Samples from vaccinated patients were analyzed to verify the presence of anti-spike antibodies. By these criteria, 8 samples from the infected groups were excluded and 16 samples from the vaccinated groups were excluded from analysis, resulting in 195 positive samples. All patients consented to the study.

**Immuno-assay procedure**: Lateral flow assays (LFAs): The following nine commercial lateral flow assays (LFAs) for the detection of SARS-CoV-2 specific IgM and IgG were analyzed in this study (Table 2): Abnova (COVID-19 Human IgM/IgG Rapid Test; Abnova Co.; Taipei, Taiwan); Nadal (Nadal COVID-19 IgG/IgM test; Nal Von Minden GmbH; Moers, Germany); Ring Biotech (COVID-19 IgM/IgG Rapid Test Kit; Ring Biotechnology Co.; Beijing, China); Wondfo (SARS-CoV-2 Antibody Test; Guangzhou Wondfo Biotech Co.; Guangzhou, China); Labnovation (COVID-19 IgM/IgG Antibody Test Kit; Labnovation Technologies; Shenzhen, China); CTK (OnSite COVID-19 IgG/IgM Rapid Test; CTK Biotech; Poway, CA, USA); Biosynex (Biosynex COVID-19 BSS (IgG/IgM); Biosynex Swiss SA; Delémont, Switzerland); Dynamiker (2019-nCoV IgG/IgM Rapid Test; Dynamiker Biotechnology; Tianjin, China); Cortez (SARS-CoV-2 Antibody (IgG/IgM); Diagnostic Automation/Cortez Diagnostic; Woodland Hill, CA, USA).

All LFAs were performed in accordance with the manufacturer’s guidelines. In summary, reagents were brought to room temperature, patient sera were completely thawed prior to testing, and 10 μL of the sample was pipetted into the sample well, followed by the addition of a sample buffer. Cassettes were read within the specified time window and photographed to document the result. Tests were considered valid if the control band was present. As this is a subjective reading, we distinguished between a positive test (clearly visible test band) and a weakly positive test (slightly visible test band). According to the instructions of the suppliers, both results are considered positive in our analysis.

ELISA and CMIAassays: Samples were tested to detect anti-spike IgG (Anti-SARS-CoV-2 ELISA kit; Euroimmun), anti-nucleocapsid IgG (SARS-CoV-2 IgG chemiluminescent microparticle immunoassay; Abbott). The analyses were performed following the manufacturer’s instructions (the reaction volumes were 10 μL for the ELISA test and 25 μL for the CLIA test). The reported sensitivity is 96% (ELISA) and 94% (CMIA) for samples from patients with severe infection collected >14 days post-onset, and the reported specificity is 99.8% (ELISA) and 100% (CMIA) [9].

**Statistical analysis**: A power calculation (alpha 5%, power 90%) based on expected sensitivity (85%, Whitman et al.) was used to determine the minimum sample size required to assess the sensitivity of several serological tests for COVID-19, using expected binomial exact 95% confidence limits, resulting in an estimated number of positive cases of 45. All the test results were analyzed using R [12,13,14,15,16,17]. Normal regression models were built using the “glm” function with the Gaussian family and identity link. Analysis of variance was then used to validate the results of each model. Each test, vaccine, and immunity type was evaluated separately. This effectively allowed an evaluation of the performance of each test and the measurement of the impact of the relevant variables—namely, gender, age, and time difference between vaccine or infection and test. The results of each model are summarized by displaying significant contributing variables (“Pr(>|t|)” and “Pr(>Chi)” < 0.005).

## 3. Results

Reliable LFAs must meet both specificity and sensitivity criteria. Specificity was determined with 69 control samples, all of which were determined to have negative anti-spike IgG by Euroimmun ELISA or negative anti-nucleocapsid IgG by Abbot immunoassay. Six of the nine assays had a specificity >98% (Table 3).

The sensitivity of the LFAs was determined with 195 positive samples and ranged from 9.0 to 90.3%. Three assays had sensitivities above 80%, but none fulfilled the criterium of >95%. Discriminating the groups in post-infection and post-vaccination showed that sensitivities ranged from 4.6–93.6% (Table 3). Two assays had a mean sensitivity above 80% and the confidence interval of one assay included the minimum criterium of 95% (Cortez). Samples from 101 vaccinated people were used to determine the sensitivity of vaccination responses. The sensitivity of the assays for post-vaccination responses ranged from 14.0 to 87.1%. None of the assays met the minimum criterium for sensitivity when post-vaccination samples were used.

Vaccination responses were divided by response after mRNA vaccine versus inactivated virus vaccine, since after mRNA vaccine only antibodies to the spike protein will be generated (Table 3). The LFA instructions from Ring Biotech, Wondfo, Dynamiker, and Cortez stated to use the nucleocapsid protein as an antigen, so sensitivity for post-vaccination response after mRNA vaccination was expected to be absent or at least much lower than the sensitivity for the post-infection response. This was indeed seen for Ring Biotech and Dynamiker LFAs, but not for Wondfo and Cortez (Table 3). Overall, the sensitivities of the LFAs containing exclusively spike protein for post-infection or post-vaccination response were comparable. Dividing the post-vaccination group according to the type of vaccine used, the mRNA vaccine group showed higher responses with the spike-antigen-containing LFAs, with the exception of the LFAs from Cortez. The samples from people vaccinated with inactivated virus vaccine showed similar responses compared to the post-infection samples, regardless of the antigenic target, with the exception of the Ring Biotech LFA. This assay showed a higher sensitivity for post-infection sera in comparison to post-inactivated vaccine sera. The reactivity seen with the nucleocapsid containing LFA in the post-mRNA vaccine group was partly due to samples from subjects that must have had SARS-CoV-2 infection, either symptomatic or asymptomatic. Five sera in this group contained anti-nucleocapsid antibodies in addition to anti-spike antibodies.

The highest sensitivities were found for spike-protein-containing assays with post mRNA vaccine samples. The confidence interval for the sensitivity of two assays (Nadal, CTK) included 95%, and for two more assays, it was almost as large (94.7%, Labnovation, Biosynex).

LFAs responses were plotted against the anti-spike IgG levels (Figure 1). If any correlation existed between the concentration of IgG and LFA results, we would expected to see an increased response of anti-spike IgG when changing from negative to positive. The LFAs of Biosynex, CTK, Labnovation, and Nadal showed results reflecting that these LFAs used spike protein as an antigenic target. However, the test results from Abnova, Ring Biotech, and Dynamiker did not show positive LFAs responses in samples with high anti-spike IgG levels, reflecting the fact that these LFAs use nucleocapsid protein as an antigenic target. The Wondfo and Cortez LFAs should contain spike protein aside from the nucleocapsid protein, since a correlation was observed between the semi-quantitative LFAs response and the anti-spike IgG titer.

Displaying the data per vaccine, it becomes evident that, except for the Abnova test, all LFAs showed a clear response correlation with the anti-spike IgG response after vaccination using a virus inactivated vaccine (Sinovac). As expected, IgG generated with the mRNA vaccines (Pfizer, Moderna) gave positive responses in spike-protein-containing LFAs and negative responses in nucleocapsid-containing LFAs (Figure 2).

These similarities and differences are more obvious when observing the correlation matrices. With these correlation matrices, it appears that Ring Biotech, Abnova, and Dynamiker all use nucleocapsid targets exclusively, and the others target at least the spike protein (Figure 3A). The second correlation matrix (Figure 3B) illustrates the relative performances of the tests, showing Abnova as a clear under-performer. Dynamiker and Ring Biotech show very similar results, both exclusively using nucleocapsid as an antigen. The LFAs with only spike protein as an antigenic target cluster with a high correlation. Cortez clusters with the spike-protein-containing assays, although the insert states that the test has nucleocapsid as an antigenic target. Wondfo appears different from the other tests, which is consistent with the observed lower sensitivity for post-infected samples in comparison to the other tests. Figure 3B seems to indicate that, overall, the tests with the same characteristics show globally similar performances. Therefore, it seems that if one knows precisely which antigens are present on the lateral flow device, one could predict the performance of the test.

Supporting these findings, linear model results are presented in the Appendix A, and a summary of these results is presented in the Appendix A. These linear models allowed us to test whether there is a linear response between different variables and the LFA results. From those results, it is clear that anti-spike IgG levels linearly correlate with the LFA results when these target the spike protein. Additionally, these linear model results clearly show no effect of gender, date between sampling and infection/vaccination, or age (Appendix A).

## 4. Discussion

Government guidelines should always be consulted when selecting appropriate LFAs. In addition, independent assessment with predefined conditions applied to a wide selection of kits allows a comprehensive insight into the performance of these kits. In this study, except for the first cohort, for which we had confirmatory PCR, we based our positive/negative selection of all samples with best-in-class quantitative serological assays in a clinically certified environment. Since these assays are known to have very high specificity [18,19], the chance of wrongly defining a sample as a false positive is excluded. The sensitivity of the used immunoassays is also high, but not 100%. Using the definition of two positive results to identify a sample as positive may have resulted in the exclusion of a few positive samples. However, the positive criteria were stringent to exclude response uncertainty. Evidence that the stringency was correctly defined was confirmed by the fact that not all the samples from the first cohort, whilst PCR-positive, met the positive criteria as defined by the authors.

Our assessment succeeded in the classification of LFAs kits in performance and guided our choice for the most performant kit in terms of sensitivity and specificity, independent of the LFAs kit manufacturer’s documentation. There is not a lot of literature about the minimum performance criteria of serological SARS-CoV-2 assays. Van Walle et al. [5], in their meta-analysis of clinical performance of commercial SARS-CoV2 nucleic acid and antibody tests, collected performance criteria that are proclaimed in Europe. Only two minimum criteria regarding serological assays were found: one from the Dutch National Institute for Public Health and the Environment stating that the minimal sensitivity for IgG assays has to be 95% and the minimum specificity 98%, the second from the Medicines and Healthcare Products Regulatory Agency from the United Kingdom claiming a minimum of 98% (96–100% CI) for both sensitivity and specificity [5]. We applied the least stringent Dutch criteria. Six assays met the criteria (>98%) for specificity and none met the criteria for sensitivity (>95%) using post-infection and post-vaccination serum samples. Subdividing the sensitivities according to the status—i.e., either post-infection or post-vaccination (mRNA vs. inactivated virus)—showed that some assays barely reached the minimum criteria for sensitivity for a selected subgroup: Cortez for post-infection and post-inactivated virus vaccine, Nadal and CTK for post mRNA vaccine samples. This reflects again the relation between the test result and the antigenic target.

Although all LFAs were able to detect both IgG and IgM, for this study we only included the IgG responses for higher assay robustness. All positive samples had been shown to have positive IgG responses against spike protein (post-vaccine samples) and both spike and nucleocapsid (post-infection) in the immuno-assays. In addition, most samples were taken >20 days after infection or vaccination, so an analysis of the IgM response would not be of additional value.

Based on the results obtained with the carefully composed study cohort containing post-infection samples, as well as samples from either mRNA-vaccinated or inactivated-virus-vaccinated people, we could cluster the LFAs with regard to the responses seen. Both post-infection and samples from subjects vaccinated with the inactivated virus are expected to contain anti-nucleocapsid antibodies in addition to the anti-spike antibodies, while sera will contain only anti-spike antibodies after mRNA vaccination. For six assays, we had a claim from the manufacturer about the antigenic target. Two out of six were not correct, in such that apart from the claim that the antigenic target was nucleocapsid protein, spike protein had to be present as well, since we found a high correlation with the semi-quantitative LFA response and the anti-spike IgG response and the heat map clustering of these assays with the assays containing spike protein exclusively. Two of the three assays of which we did not receive any information about the antigenic target contained spike protein according to our heatmap analysis. The third assay with an unknown antigenic target performed very poorly, with sensitivities of less than 20% in all groups. The fact that this assay met the specificity criterium probably reflects the low reactivity of the test system and should not be considered as a pass.

In general, the sensitivities of the spike-antigen-containing LFAs were highest when analyzing post-mRNA vaccine samples, reflecting the significantly higher anti-spike IgG responses in these samples compared to the other groups. It cannot be excluded that subjects in the post-mRNA vaccine group may have had a SARS-CoV-2 infection in the past, either symptomatic or asymptomatic. We demonstrated anti-nucleocapsid antibodies in five samples besides the anti-spike antibodies, which must have been induced by infection. However, these samples did not have higher anti-spike IgG titers compared to samples in this group that did not have anti-nucleocapsid antibodies, so it is unlikely that a past infection influenced the results.

We did not find any effect of gender, date between sampling and infection/vaccination, or age. However, the different groups were not evenly balanced: we had twice as many females as males, infected people were all under the age of 60, and only the mRNA-vaccinated group contained people over 60.

The difficult time of the pandemic induced the emergence in the market of a multitude of available tests with variable specifications [6,20], some with poor performance. The claims of the performance characteristics by the manufacturer are often too promising or are missing entirely. Kierkegaard et al. systematically studied the supportive information of SARS-CoV-2 point-of-care (POCT) devices and concluded that commercial manufacturers need to improve the quality of the information they provide for POCTs [20]. By using a carefully designed panel to verify SARS-CoV-2 LFAs, we demonstrated the inconsistency of the information given regarding the antigenic target. We also showed that none of the nine tests fulfilled the performance criterium regarding the sensitivity, regardless of their intended use.

Despite the technical limitations that we have highlighted in our study, we believe that there may be possible applications for these devices. Studies suggest that monitoring antibody concentrations can be used to optimize vaccination strategies and monitoring of individual patients (especially the most vulnerable) by estimating antibody levels [21]. The protection thresholds vary according to the studies and, at present, there is not yet a consensus [22]. An additional analysis of our data shows that above an index of 3 with the anti-spike IgG ELISA from Euroimmun (equivalent to around 150 BAU/mL), the sensitivity of certain tests meets the sensitivity criterion, and we could imagine using them for this type of monitoring. Of course, the use of a quantitative test is preferable, but the use of LFAs could be deployed where more sophisticated tests are not available, for technical, logistical, or economic reasons. Additionally, LFAs have also been developed for use with capillary blood, which allows their use as POCTs. This gives an advantage to these tests in situations where the collection of venous blood, the storage of samples, and other logistical aspects are a hindrance to these analyses. The choice of test should be appropriate for the types of vaccines used locally. Additional studies on each device are nevertheless necessary, but we believe that the data from this study are a useful basis for reflection.

## Figures and Tables

**Figure 1 jcm-11-02100-f001:**
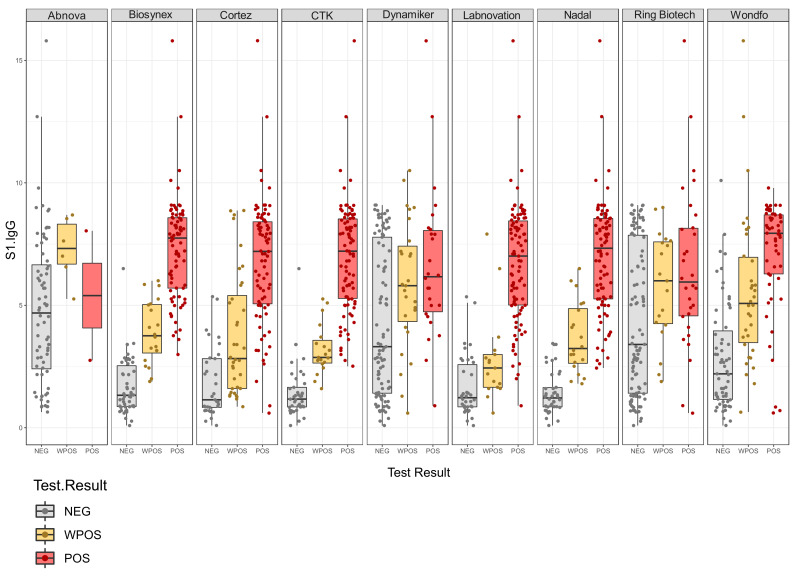
Plotted are the semi-quantitative LFA results (*x*-axis) against the anti-spike IgG response (*y*-axis). Results are categorized as negative IgG response (NEG), weakly positive IgG response (WPOS), and positive IgG response (POS).

**Figure 2 jcm-11-02100-f002:**
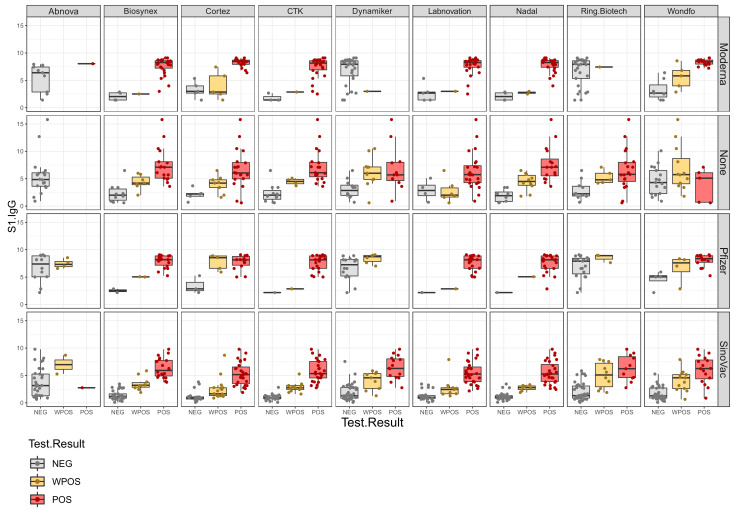
Semiquantitative LFA responses (*x*-axis) versus anti-spike IgG response (*y*-axis) sorted by vaccine (Moderna and Pfizer represent mRNA vaccine, SinoVac represents inactivated virus vaccine, and None represents the post-infection cohort). Results are categorized as negative IgG response (NEG), weak positive IgG response (WPOS), and positive IgG response (POS).

**Figure 3 jcm-11-02100-f003:**
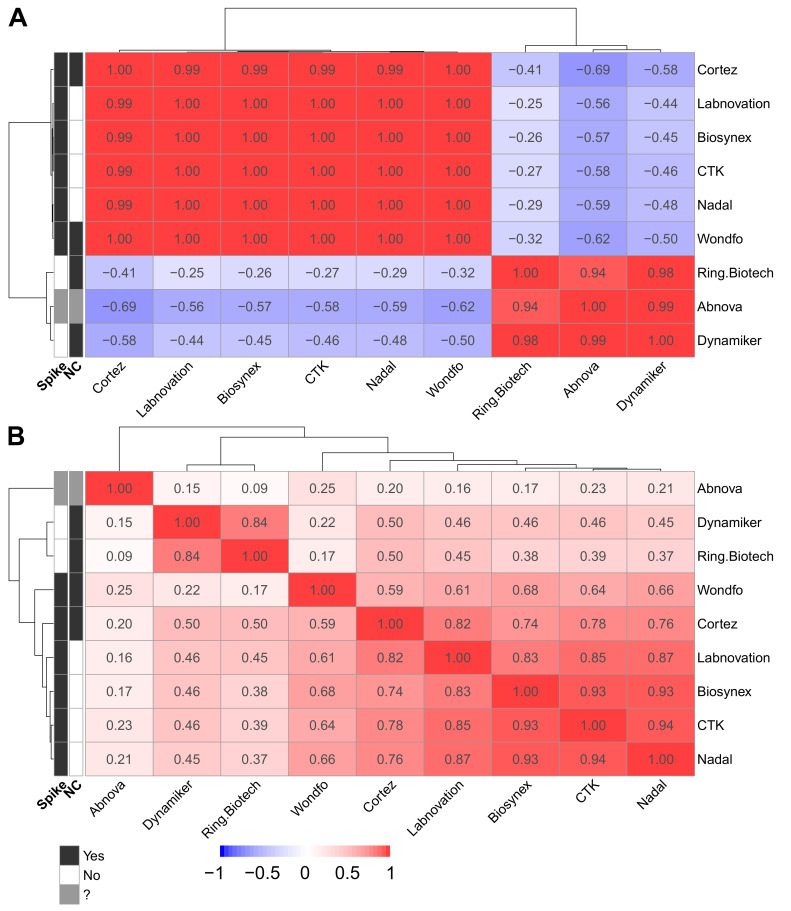
(**A**) Correlation heat map of anti-spike IgG and LFA results sorted by LFA and antigen. Clustering of spike-protein-containing LFAs and exclusively nucleocapsid-containing LFAs can be seen. The LFA from Abnova clusters with the nucleocapsid protein-containing assays. (**B**) Correlation heat map of anti-spike IgG and LFA results sorted by LFA. The output of the regression models used can be found in the Appendix A.

**Table 1 jcm-11-02100-t001:** Characteristics of the sample cohorts.

Cohort (n =)	Female/Male	Mean Age (Range)	Positive *
Post-infection (35)	29/6	41 (22–58)	30
Health personnel (101)	52/17	53 (18–91)	32
Seroprevalence study (35)	27/8	43 (17–78)	32
mRNA vaccinated (52)	34/18	53 (25–84)	5
Inactivated virus vaccinated (64)	35/29	36 (922–56)	18

* The number of samples that were positive for both anti-spike IgG and anti-nucleocapsid IgG. For the mRNA vaccinated group, 47 samples contained anti-spike IgG only.

**Table 2 jcm-11-02100-t002:** Lateral flow assays with manufacturer’s specifications: the manufacturer, batch number, serum volume, sample buffer, and incubation time are listed. Target antigens are listed when available.

Company	Batch Number	Antigen (*)	Serum	Sample Buffer	Incubation
Abnova	K4081	not specified	10 μL	100 μL buffer	15′
Nadal	COV20040036	not specified	10 μL	2 drops	10′
Ring Biotech	20200325	N	10 μL	4 drops	10–15′
Wondfo	W195004104	N	10 μL	3 drops	15′
Labnovation	20200330	not specified	10 μL	2 drops	15–20′
CTK	F0417R8B02V	S	10 μL	2 drops	10–15′
Biosynex	COV20030128	S	10 μL	2 drops	10′
Dynamiker	200503	N	10 μL	2 drops	10′
Cortez	C05250	N	10 μL	2 drops	15′

* N = nucleocapsid protein and S = spike protein.

**Table 3 jcm-11-02100-t003:** Sensitivity of lateral flow assays with manufacturer announced target and observed target. The sensitivity was calculated for post-COVID-19 and post-vaccination patients (vaccinated with mRNA and inactivated virus vaccines).

LFA	Target	Target Observed	Sensitivity (%) Overall	Sensitivity (%) Post-COVID	Sensitivity (%) Post Vaccination	Sensitivity (%) Post mRNA Vaccine	Sensitivity (%) Post Inactivated Virus Vaccine	Specificity (%)
Abnova			9.0 (5.1–15.4)	4.6 (1.6–12.7)	14.0 (7.3–25.3)	18.5 (8.2–36.7)	10 (3.5–25.6)	100
Nadal		S	79.0 (72.7–84.1)	75.5 (66.0–83.1)	82.2 (73.6–88.4)	90.6 (79.7–95.9)	72.9 (59.0–83.4)	100
Ring Biotech	N	N	53.3 (46.3–60.2)	84.0 (75.3–90.1)	24.8 (17.4–34.0)	7.5 (3.0–17.9)	43.8 (30.7–57.7)	97.1
Wondfo	N	S, N	58.2 (51.2–65.0)	48.4 (38.5–58.4)	67.3 (57.7–75.7)	77.4 (64.5–86.5)	56.3 (42.3–69.3)	98.6
Labnovation		S	84.6 (78.9–89.0)	87.2 (79.0–92.5)	82.2 (73.6–88.4)	88.7 (77.4–94.7)	75.0 (61.2–85.1)	94.2
CTK	S	S	80.5 (74.4–85.5)	74.5 (64.8–82.2)	86.1 (78.1–91.6)	92.5 (82.1–97.0)	79.2 (65.7–88.3)	97.1
Biosynex	S	S	73.8 (67.3–79.5)	71.3 (61.4–79.4)	76.2 (67.1–83.5)	88.7 (77.4–94.7)	62.5 (48.4–74.8)	100
Dynamiker	N	N	48.2 (41.3–55.2)	66.0 (55.9–74.7)	31.7 (23.4–41.3)	15.1 (7.9–27.1)	50.0 (36.4–63.6)	98.6
Cortez	N	S, N	90.3 (85.3–93.7)	93.6 (86.8–97.0)	87.1 (79.2–92.3)	84.9 (72.9–92.1)	89.6 (77.8–95.5)	98.6

## Data Availability

Data supporting the reported result can be accessed by corresponding with the authors.

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
