# Peer review of "A Comparative Study of Nine SARS-CoV-2 IgG Lateral Flow Assays Using Both Post-Infection and Post-Vaccination Samples"

_jcm, 2022, doi:10.3390/jcm11082100_

Round 1
Reviewer 1 Report
This study has attempted to compare the sensitivity and specificity of different commercially available lateral flow assays. The following concerns should be addressed to make the findings from this study stronger:
- Lines 98-106: Were the samples blinded before testing using different LFAs to reduce bias in the findings? More information (gender, age, etc.) should be provided on the cohorts that the sera samples were obtained from.
- Lines 114-117: More details on the ELISA protocol should be provided. For example, what volume of sera were used for the assays? What is the specificity and sensitivity of the ELISA kits? What assay controls were used for the ELISA?
- Line 111 and Table 1: Does 10 l mean 10 ul?
- Figures 1 and 2: The legends for the figures need improvement. A description of NEG, WPOS and POS should be provided in the legend. In addition, the axis numbers and other labeling should be increased in size to improve legibility
- Were LFA responses compared to anti-N IgG levels for LFAs that are specific to nucleocapsid only?
- Lines 212-213: Data showing these findings should be presented in the manuscript
Author Response
Thank you very much for your comments, which have been very useful to us.
Below you will find our comments and information regarding the changes we have made to the text:
- We have adapted the text and added a summary table with the information you requested (table 1). We did not follow a sample anonymization protocol, but the laboratory personnel who performed the LFAs did not have access to the ELISA and CMIA test results.
(Changes to lines 108-115)
- We have added in the text the performance data of the ELISA and CLIA tests, as well as the reaction volumes (in order to have a comparison with the LFA). We have not mentioned other technical characteristics, because they do not seem to us to have a significant effect on the results of these two tests.
(Changes to lines 137-144)
- It is indeed 10 m It seems that there was a problem with the submitted text and we will make sure that in the next version this problem is not present.
(Changes to line 130)
- We have taken this remark into account and modified the document both at the level of the figures and of the “Materials and Methods” section. Explanation of the concept of a weakly positive test: as it is a subjective reading, we have distinguished between a positive test (clearly visible test band) and a weakly positive test (inconspicuous test band). According to the instructions of the suppliers, both results are considered positive in our analysis. Nevertheless, we believe that this information is useful when choosing one of the tests by a reader of the article.
(Changes to lines 133-136, 220-222, 232-236)
- We analyzed the tests to see if they were able to detect and discriminate patients as uninfected, infected or vaccinated. It is clear that LFAs coated with nucleocapsid protein that do not detect patients vaccinated with mRNA vaccines. Moreover, only 18 out of 64 sera from patients vaccinated with a vaccine containing an inactivated virus showed the presence of anti-Nucleocapsid antibodies. We wanted to limit our analysis to the data presented and to remain at the level of the patient's serological status (non-infected, infected and vaccinated). Indeed, it seems to us that a more detailed analysis at the level of anti-Nucleocapsid antibodies would have lengthened the text, without however providing any real significant additional information.
(No changes in the manuscript)
- We added in manuscript a comment concerning figure 3B. Theoretically, this would be more of a discussion than an analysis of the results. Nevertheless, we have added the two sentences in the section, to keep the reading flowing.
(Changes to lines 248-251, 206-215, 260-266)
Reviewer 2 Report
I was informed by the journal that enough review reports were sent, so mine is not needed. Therefore, I’ll keep it short, only mentioning a few points that need to be discussed/addressed.
The study is certainly of interested to the reader.
- The authors should give more details, how the tests were chosen. Why these nine LFAs? It is of interested to the reader -> are they broadly available, cover a broad range of analytes etc.?
- The introduction mentions the uses cases for serological SARS-CoV-2 assays. The authors need to give more details here, or rephrase the sentences. As far as I know, there are no titer cut-offs for protection etc. Are there any recommendations for treatment etc. that are based on a titer? Otherwise, the titers are more or less just a measure for exposure?
- I think an approval from an ethics comittee might be needed. As far as I know, it's not enough that material is anonymized, an votum is needed whenever new experiments are carried out with patient material. At least at our institute the committee has to make sure that the use of leftover specimen is in line with ethical principles. So if these LFA results were not obtained before for other reasons and analyzed for this study retrospectively, I believe such a votum is needed.
Author Response
Thank you very much for your comments, which have been very useful to us.
Below you will find our comments and information regarding the changes we have made to the text:
- We added to the manuscript the mode of selection of the tests (“We selected four tests available in Europe and previously evaluated by one of the authors (Dr Giuseppe Togni) for its activity in the company Unilabs SA (Switzerland) and five selected from a large panel of available tests”).
(Changes to lines 87-89)
- We have modified the paragraph. The message is no longer based on how serology could be used in a number of situations that we have mentioned. It emphasizes the importance of having a wide range of reliable analytical tools to meet all epidemiological, diagnostic, logistical and economic scenarios.
For your information, in certain circumstances, attempts have already been made to set decision thresholds. For example, in Switzerland, the competent authorities had decided that serological results should be used to issue certificates of cure and they recommended using them for the vaccination management of immunocompromised patients.
(Changes to lines 57-60)
- We would like to provide the following rationale for the approach taken on ethical approval. The study was carried out with five panels of samples from three different countries (Switzerland, Holland and Peru). Three of the five panels had been approved by an ethics committee to use of samples for epidemiological purposes (Switzerland and Holland). The fourth panel concerns samples of health care workers who had given their written informed consent (Peru). Finally, the last panel concerns samples from health care workers and patients who had also given their written informed consent (Switzerland).
The study has evolved over time. First, we analyzed samples from infected and uninfected patients. With the arrival of vaccines, we decided to also include patients vaccinated with two different types of vaccines as an additional factor for consideration. In our opinion, this is make the study more interesting and instructive.
With this addition, we understand that there could an implication from a regulatory point of view, but we are nevertheless confident the minimum requirements are met. Indeed, all the participants gave their written informed consent and two ethics committees have approved the use of the samples for epidemiological purposes.
The main reason in our opinion as to why a request to an ethics committee was not required is the following. This study had no clinical and diagnostic impact for patients. Indeed, no clinical or therapeutic management decision were made. Thus, this study can be compared to a retrospective evaluation of methods, which does not necessarily require approval from an ethics committee.
(No changes in the manuscript)
Reviewer 3 Report
the study by Mulder L and colleagues evaluates the sensitivity and specificityof 9 different lateral flow assays to identify the presence of antibodies
against SARS-CoV-2 viral proteins in the serum of people with previous
infection, of vaccinated subjects and of a group control. the study presents
points of interest, such as the absolute divergence between the results
obtained and the values ​​declared by the producers, but at the same time
it presents weaknesses. The number of subjects enrolled is low
. As regards the group of vaccinated subjects, being the diagnosis based
on serology (ELISA), we cannot in any way exclude any previous asymptomatic
infection (very common in subjects with COVID-19). The latter is not a
negligible effect considering that a lateral flow test in an unvaccinated
community of massachusetts showed a high positivity to antibodies (an index
of asymptomatic previous infection) REF: Naranbhai V, et al. High
Seroprevalence of Anti-SARS-CoV-2 Antibodies in Chelsea, Massachusetts.
J Infect Dis. 2020;222:1955-9. doi: 10.1093/infdis/jiaa579. Also, although
never all 9 together, there are many other studies that have evaluated the
accuracy of the different lateral flow essays. For example, Nadal was
evaluated in this study. REF: krone M, et al. Performance of Three SARS-CoV-2
Immunoassays, Three Rapid Lateral Flow Tests, and a Novel Bead-Based
Affinity Surrogate Test for the Detection of SARS-CoV-2 Antibodies in
Human Serum. J Clin Microbiol. 2021;59(8):e0031921. doi: 10.1128/JCM.00319-21.
Authors should improve their review of the existing literature on this same
topic. The authors should also improve the definition of figures 1 and 2.
In fact, the low resolution used prevents them from being read. Tables should
be formatted properly and text spacing is often double (and should be
corrected).
Author Response
Thank you very much for your comments, which have been very useful to us.
Below you will find our comments and information regarding the changes we have made to the text:
- “The number of subjects enrolled is low”
We are aware that a higher number of participants would have had a beneficial effect on the statistical analysis. However, in similar studies the number of patient is similar (ref. 7, Krone et al,2021, J Clin Microbiol : 63 patients ; ref. 8, Mylemans et al,2021, J Immunol Methods : 162 patients).
One of the authors (Dr Giuseppe Togni) recently published an article in your journal on the comparison of automated serological tests (Andrey et al , 2021, J Clin Med 2021) where the number of patients was 357. As mentioned in the manuscript, the minimum number of positive samples needed to perform a reliable statistical analysis of the sensitivity of the tests was calculated. In our panel there were more than the 45 samples required and we consider that the number of patients is sufficient.
(No changes in the manuscript)
- « As regards the group of vaccinated subjects, being the diagnosis based
on serology (ELISA), we cannot in any way exclude any previous asymptomatic
infection (very common in subjects with COVID-19).”
In the panel of patients vaccinated with mRNA vaccinia, five samples out of 52 showed the presence of anti-Nucleocapsid antibodies, indicating that there had also been contact with the virus. However, these patients did not have significantly higher anti-Spike antibody titers than other vaccinated patients. This observation was mentioned in the “discussion” section of the manuscript.
(Changes to lines 199-202, 323-329)
- As requested, we have added complementary references, including the one proposed (ref. 7, Krone et al, 2021, J Clin Microbiol) and we thank you for that.
(Changes to lines 76-85)
- We have tried to improve the quality of tables and figures to make them more readable and understandable.
(Changes in table 3, figures 1 and 2)
Round 2
Reviewer 2 Report
I am still unsure if the authors would need one for this study, therefore I am not able to judge, whether the paper can be published or not. I'd advise the journal or the authors to get a statement from an/their ethics committee stating that indeed no votum is needed for this study before publication.
Reviewer 3 Report
I really appreciate the efforts made by the authors to improve their article. However, I do not believe that it reaches such a priority as to allow its publication on JCM.